# Effect on an Oral Nutritional Supplement with β-Hydroxy-β-methylbutyrate and Vitamin D on Morphofunctional Aspects, Body Composition, and Phase Angle in Malnourished Patients

**DOI:** 10.3390/nu13124355

**Published:** 2021-12-03

**Authors:** Isabel Cornejo-Pareja, Maria Ramirez, Maria Camprubi-Robles, Ricardo Rueda, Isabel Maria Vegas-Aguilar, Jose Manuel Garcia-Almeida

**Affiliations:** 1Department of Endocrinology and Nutrition, Virgen de la Victoria Hospital (IBIMA), Malaga University, 29010 Malaga, Spain; isabel.mva13@gmail.com; 2Centro de Investigacion Biomedica en Red de la Fisiopatología de la Obesidad y Nutricion (CIBEROBN), Instituto de Salud Carlos III (ISCIII), 29010 Malaga, Spain; 3Abbott Nutrition, Research and Development, 18004 Granada, Spain; maria.ramirez@abbott.com (M.R.); maria.camprubirobles@abbott.com (M.C.-R.); ricardo.rueda@abbott.com (R.R.)

**Keywords:** malnutrition, muscle loss, oral nutritional supplements, cancer, β-hydroxy-β-methylbutyrate, phase angle, body cell mass

## Abstract

This is a retrospective study of data from clinical practice to observe the effect of a high-calorie, high-protein oral nutritional supplement (ONS) with β-hydroxy-β-methylbutyrate (HMB) on nutritional status, body weight, and muscle-related parameters in 283 adult patients with or at risk of malnutrition under standard of care, 63% being cancer patients. They were recommended to increase physical activity and energy and protein intake from regular diet plus two servings per day of a specialized ONS enriched with HMB or standard ONS for up to 6 months. Dietary records, adherence and tolerance to ONS, nutritional status, body composition, handgrip strength, and blood analysis at the beginning and the end of the intervention were recorded. This program improved nutritional status from 100% malnourished or at risk of malnutrition at baseline to 80% well-nourished at final visit. It also increased body weight by 3.6–3.8 kg, fat-free mass by 0.9 to 1.3 kg, and handgrip strength by 4.7 to 6.2 kg. In a subgroup of patients (*n* = 43), phase angle (PhA), and body cell mass (BCM) increased only in the patients receiving the ONS enriched with HMB (0.95 (0.13) vs. −0.36 (0.4), and 2.98 (0.5) vs. −0.6 (1.5) kg, mean difference (SE) from baseline for PhA and BCM, respectively), suggesting the potential efficacy of this supplement on muscle health.

## 1. Introduction

Malnutrition is an increasingly prevalent condition in healthcare, which has a profound impact on patients’ clinical and functional outcomes. Prevalence of malnutrition ranges between 30 and 50% in hospitalized adult patients and 30–60% in older adults from nursing homes [1]. Malnutrition results in diminished physical and mental function [2]. One of the main consequences of malnutrition is muscle mass and strength loss, which is associated with impaired outcomes such as mobility-disability, illness, increased risk of infections, poor quality of life, and mortality [3].

In a recent study, Pourhassan et al. [4] demonstrated that malnutrition according to the Global Leadership Initiative on Malnutrition (GLIM) criteria was significantly and independently associated with acute muscle wasting measured by magnetic resonance imaging (MRI) in frail older patients during 14 day hospitalization.

These findings are remarkable and suggest that, ideally, identifying and treating malnutrition should have a focus on trying to maintain or prevent muscle mass and function loss [5].

However, the screening and assessment of muscle loss may be a significant challenge for clinicians, as this strongly depends on the use of validated and sometimes expensive tools, which are not always available in clinical practice.

The incorporation of muscle mass-loss assessment as an essential phenotypic criterion into the recent GLIM criteria for the diagnosis of disease-related malnutrition (DRM) [6], makes it necessary to systematically implement morphofunctional tools to assess muscle health. Although there is no universally validated technique for the assessment of reduced muscle mass, it is important to highlight the usefulness of standard tools such as bioelectrical impedance analysis (BIA) that can be easily implemented in routine clinical practice [6]. Likewise, there are proposed cut-off points to evaluate the diagnosis of muscle mass loss through parameters such as FFMI (fat-free mass index), SMI (skeletal muscle index), and ASMM (appendicular skeletal muscle mass) [7,8].

The traditional standard tool, BIA, estimates body composition indirectly based on predictive equations. Bioelectrical impedance vector analysis (BIVA) provides raw electrical values such as phase angle (PhA), which has been demonstrated to be a good predictor of muscle abnormalities and function with a good diagnostic accuracy for detecting low muscle mass in cancer patients [9]. Importantly, its direct application may predict body composition changes, making PhA a good marker of nutritional status as well as a risk indicator for morbidity and mortality [10].

Reduced muscle mass and function has important health implications, especially in patients with disease or conditions with inflammatory components such as chronic obstructive pulmonary disease, chronic kidney disease, and cancer [11]. Muscle loss is especially relevant in cancer-related malnutrition, as both the tumor and the anti-cancer treatment can negatively impact the patient’s nutritional status and muscle health. Several studies in Europe have reported that prevalence of malnutrition in cancer patients can be up to 31% or even up to 83% in older adult populations [12,13,14]. Indeed, almost 20% of cancer patients die because of malnutrition and its complications rather than the malignancy itself [15].

Chronic diseases and inflammatory conditions systematically lead to muscle atrophy mainly due to altered protein synthesis, altered metabolism, and increased energy requirements. In particular, muscle mass loss in cancer patients is associated with increased treatment toxicity and reduced tolerance to anti-cancer treatment [16,17].

Malnutrition and associated muscle loss are not inevitable, as addressing the modifiable lifestyle factors of diet and physical activity has been proven to be the most effective strategy for prevention and treatment in clinical practice. Recently, in a large study recruiting 2088 patients, the Effect of early nutritional support on Frailty, Functional Outcomes, and Recovery of malnourished medical inpatients Trial (EFFORT) showed that nutritional support was able to effectively counteract negative outcomes in hospitalized patients by increasing caloric and protein intake, which led to improved survival rates compared to standard hospital food [18].

Nutritional support can reverse malnutrition and muscle loss. Different ingredients have been proposed to play an important role in improving nutritional status as well as preserving muscle health. Protein is one of the most effective anabolic stimuli to support muscle health. Current European Society for Parenteral and Enteral Nutrition (ESPEN) guidelines recommend protein intake between 1.2 and 1.5 g/kg of body weight (BW)/day for certain older adults who have acute or chronic illnesses [19].

Another important anabolic nutrient is β-hydroxy-β-methylbutyrate (HMB), which has been recognized in ESPEN guidelines as an important functional ingredient to maintain muscle mass in cancer patients [20].

Therefore, this evidence may suggest that reversing low muscle mass and function is feasible by using nutritional intervention and has the potential to improve clinical outcomes in patients with or at risk of malnutrition under different disease settings.

Herein, the aim of this study was to observe the effect of a high-calorie, high-protein oral nutritional supplement (ONS) with HMB on nutritional status, body weight, and muscle-related outcomes in adult patients with or at risk of malnutrition under standard of care.

## 2. Materials and Methods

### 2.1. Study Design

This study is a retrospective analysis of a database that was built at the Virgen de la Victoria Hospital, Malaga, Spain from April to December 2017. The patients attended consecutively to a specific nutrition medical office because they were diagnosed with malnutrition and they needed nutritional treatment. An important proportion of them were oncological patients (*n* = 179). They were attended according to the clinical practice of the department, and the retrospective study of their data was approved by the Ethics Committee of the hospital.

A trained dietician provided recommendations for diet enrichment and physical activity for muscle strengthening. Exercises aimed at muscle strengthening were recommendations established in our center together with the Rehabilitation Unit. The recommendation was strength exercises with weights and elastic bands, of progressive intensity until reaching fatigue, three times per week and with a session duration of 30–45 min. We proposed both upper-limb strength exercises (shoulder extension and row with elastic bands or shoulder holds and push-up arms with weights) and trunk and lower limb (trunk extension, abdominals, squats or leg and knee extension, knee bend, tiptoe).

The dietician insisted on a series of points in all patients for caloric and protein enrichment, such as distributing the intakes, increasing and prioritizing the consumption of meat, fish, and eggs (minimum three servings a day), increasing the consumption of carbohydrates such as bread or flours, potatoes, pasta, legumes, rice, and incorporating caloric foods in food preparation: cheeses, fats, creams, nuts, olive oil, whole yogurts, powdered milk. The patients were asked also to supplement their regular diet with two servings of an ONS, either a specialized ONS (S-ONS group), high in energy and protein, and enriched with HMB (calcium salt) or a standard ONS (ONS group) until next visit, which was between 3 and 6 months. The ONS servings were recommended one in the morning and one in the evening, one of them around the time of exercise.

Dietary recommendations and type of ONS were established individually for each patient based on their clinical needs and based on healthcare standards for malnutrition patients. The recommended calorie intake was set at 30–35 kcal/kg BW/day and protein at 1.2–1.3 g/kg BW/day. Dietary therapy, adherence, and tolerance to ONS, as well as body weight and body composition were evaluated at the beginning and the end of treatment. Both groups received the same nutritional care and physical activity recommendations.

### 2.2. Oral Nutritional Supplements

The specialized ONS contained 1.5 kcal/mL, 330 kcal energy, 20 g protein, 11 g fat and 37 g carbohydrate, 1.7 g fiber, 1.5 g calcium HMB, and 500 UI vitamin D per 220 mL serving (Ensure^®^ Plus Advance, Abbott Nutrition, Spain). The supplements that were used by the standard ONS group contained 206–320 kcal, 10.7–20 g protein, 7.8–13 g fat, and 25.5–37.7 g carbohydrates.

### 2.3. Dietary-Intake Evaluation and Clinical Data

The dietitian interviewed each patient using the 24 h recall method (for 3 days) to confirm and evaluate intake. Dietary intake was assessed based on calories using IENVA © 2010 [21]. Therefore, the adherence and tolerance to ONS were evaluated. Data regarding sex, age, diagnosis, and medical history of diseases, nutritional status, and biochemical test results were collected from electronic medical records.

### 2.4. Body-Composition Analysis

Body-composition analysis was performed using foot-to-foot bioelectrical impedance (BIA, Tanita TBF-300, Tokyo, Japan). BIA results were recorded by the dietitian at the beginning and at the end of the study. Fat-free mass (kg), total body water (kg), and fat mass (kg) were evaluated and analyzed. Height measurements were measured with a 2 mm sensitivity laser height rod.

Additionally, in a subgroup of patients, we also analyzed whole body bioimpedance measurements with a 50 kHz, phase-sensitive impedance analyzer (BIA 101 Whole Body Bioimpedance Vector Analyzer (AKERN, Florence, Italy) as previously described [22] Briefly, the method relays on the pass of an alternating current thought the body that experience a frequency dependent delay caused by cell membranes. This delay is expressed in degrees as PhA and standardized phase angle (SPhA), which is PhA adjusted by age and sex.

Other bioimpedance-derived parameters were calculated by the equipment software, namely: fat-free mass (FFM), fat mass (FM), body cell mass (BCM), total body water (TBW), extracellular water (ECW), intracellular water (ICW), appendicular skeletal muscle mass (ASMM), skeletal muscle index (SMI), hydration status, and nutrition status (creatinine excretion rate in mg/kg/24 h).

All bioimpedance measurements were obtained with the patient in supine position on a bed. Bioimpedance results were recorded by the dietitian at the beginning and at the end of the study.

### 2.5. Evaluation of Nutritional Status

Patient-generated subjective global assessments (PG-SGA) were used as a prognostic tool to evaluate the nutritional status of patients. We evaluated nutritional symptoms and weight loss. The total score was analyzed in three groups: A, well-nourished; B, moderately or suspected of being malnourished; C, severely malnourished.

Handgrip strength was determined by means of the JAMAR-Dynamometer (J A Preston Corporation, New York, NY, USA). The dominant hand was tested. Three measurements were taken, and the average was reported. Published population reference data were used as cut-off points [23].

Finally, biochemical tests were analyzed by standard laboratory methods of the hospital: prealbumin (mg/dL) (Atellica Siemens), albumin (g/dL), C reactive protein (CRP, mg/L) (Dimension EXL 200 Siemens), and CRP/prealbumin ratio.

### 2.6. Statistical Analysis

Characteristics of participants were compared in order to test for possible differences between groups. A *t*-test for independent samples comparing the S-ONS vs. ONS groups was used to compare quantitative measures and chi-square for qualitative measures. A *t*-test for paired samples with baseline and end-of-study results were used to test for differences within each treatment. A multivariate weighted regression analysis was performed to compare the S-ONS vs. ONS effect along time and adjusted by relevant clinical variables, such as age, sex, adherence, and the use of parenteral nutrition during hospital stay. For continuous variables, linear mixed models were applied with subject random effects. For categorical ordinal data, a cumulative multinomial regression mixed model was applied. Weights were obtained to balance the number of patients per group using an optimal full-matching procedure of treatment regress by the rest of covariates, as a robust alternative to propensity score weighting.

A further description and comparison of the characteristic from the groups was performed to the subset of patients that had evaluation of the bioimpedance by phase angle.

Due to the interest on assessing the influence of nutritional status on oncological patients, a further description of this subgroup is also provided.

## 3. Results

### 3.1. Baseline Characteristics

Of the 312 patients that were included in the study, 25 had missing data for intake at the end of the study and 4 did not take the supplement, so they were excluded for analysis. In total, 283 patients were analyzed, 240 took the S-ONS, and 43 received the other ONS. The baseline demographic and clinical characteristics of study participants are included in Table 1. The mean age of the patients was 61 years, with a mean body weight of 63.1 kg and BMI of 23.2 kg/m^2^. More than 60% of patients were previously admitted to hospital because of oncological surgery; the rest of the patients were admitted because of other surgical procedures or medical reasons. The oncological patients received chemotherapy or radiotherapy. Over 70% of patients were malnourished at entry and the rest of the patients were at risk of malnutrition according to PG-SGA of nutritional status. There were no differences at baseline between those who received the S-ONS or the ONS except for biceps fold and arm circumference. The S-ONS group had higher biceps fold and arm circumference. A higher proportion of patients in the S-ONS group reported acute weight loss, while in the ONS group a higher percentage of patients reported chronic weight loss (i.e., almost 30% of patients in the S-ONS group reported weight loss within 2 months before study entry vs. 17.5% in the ONS). Nevertheless, the mean percentage of weight loss in both groups was similar, 10.3 and 9.5% for the S-ONS and ONS groups, respectively, and were not significantly different (*p* = 0.561).

### 3.2. Compliance and Nutrient Intake

Over 65% of patients took more than 75% of the recommended dose, which was two servings per day. The regular diet, dietary counseling, and use of ONS significantly increased energy and protein intake in all patients. On average, before entrance to the study, they reported an intake of 1436 kcal and 55 g of protein per day (0.9 g/kg BW). With dietary counseling and the supplementation, they increased the consumption of energy and protein from the diet to 1667 kcal and 68 g of protein, and to 557 kcal and 31 g of protein from the oral nutritional supplementation. In total, they received an average intake of 2154 kcal and 97 g of protein (1.5 g/kg BW). Both groups received a similar amount of energy and protein/kg BW.

### 3.3. Effect of Oral Nutritional Supplementation

Anthropometric, body composition, functional, and laboratory outcomes at baseline and after receiving the ONS categorized by group are shown in Table 2.

There was a significant improvement in all outcomes in both groups indicating that the oral nutritional supplementation in this type of patient had a relevant effect on those measurements. The nutritional status measured by PG-SGA was significantly improved by the intervention as showed in Figure 1. At baseline, all the patients were malnourished or at risk of malnutrition, while at the end of the study about 80% were well-nourished. In fact, the probability of being well-nourished by the end of the study increased in both groups by more than 60% (*p*-value < 0.005) with no significant differences between groups.

Body weight and BMI significantly increased from baseline in both groups. On average, 3.6–3.8 kg and 1.4 kg/m^2^ were gained by both groups. The estimated increase in handgrip strength was higher in the patients receiving the S-ONS than in those receiving ONS (6.2 vs. 4.7 kg), although these differences did not reach statistical significance. FFM significantly increased from baseline in the S-ONS group but not in the ONS group (1.3 vs. 0.9 kg).

In addition, the nutritional intervention affected positively several analytical outcomes. Albumin, prealbumin, cholesterol levels, and lymphocyte counts increased from baseline to the end of the follow up period. CRP decreased three-fold from baseline. The decrease in the ratio CRP/prealbumin was significantly different from baseline only for the S-ONS group.

### 3.4. Subsample Analysis by Phase Angle

A total of 43 patients were studied by PhA: 31 were in the S-ONS group, and 12 in the ONS group.

The baseline characteristics of this subgroup were similar to those of all patients as included in Table 1, excepting that there were more oncological patients, about 70% in this subsample analysis vs. 63% of patients in the whole sample analysis. With regards to the characteristics of patients into the two subgroups, although the S-ONS patients were on average 5 years younger (65.3 ± 13.3 vs. 59.5 ± 13.8) and the percentage of oncological patients was higher than in the ONS group (22, 2, and 7 patients in the S-ONS group, and 8, 3, and 1 patients in the ONS group, were treated for oncology, general surgery, and other reasons respectively), these differences did not reach statistical significance. The daily energy intake from regular diet and from the supplement was also similar to the general group, with no significant differences between the two groups (S-ONS and ONS). With regards to protein, the S-ONS group reported on average more protein intake than the ONS group. Nevertheless, the protein intake coming from the supplement was not different (0.5 ± 0.2 and 0.4 ± 0.2 g/kg BW/day, respectively) and the total amount of protein per kg BW was within the recommendations for older adults at risk of malnutrition [19] (1.5 ± 0.4 and 1.3 ± 0.3 g for S-ONS and ONS, respectively).

Anthropometric, body composition, functional, and laboratory outcomes at baseline and after receiving the oral nutritional supplementation for the subgroups that were tested by PhA are described on Table 3. This subgroup analysis showed the same changes that were noticed for the whole sample analysis. With regards to the specific parameters obtained by BIVA, several parameters were increased from baseline in the S-ONS but not in the ONS group, namely: PhA and SPhA, BCM, ICW, nutritional status, and ASMM. A concomitant decrease in hydration percentage and ECW was also observed in the S-ONS. FFM at baseline and at final visit was higher in the S-ONS group than in the ONS group, and FFM only increased with time in the ONS group. The analysis of the differences between marginal means at the end of the study pointed out significant differences between groups in PhA, BCM, hydration status, and FFM (Figure 2).

### 3.5. Efficacy of the Specialized Oral Nutritional Supplement with HMB on Oncology Patients

One hundred and fifty-five patients received the S-ONS. At baseline, 117 patients were considered malnourished (75.5%) and 38 at risk of malnutrition (24.5%). After the program including the oral nutritional supplementation, dietary counseling, and strength exercise, 108 patients were moved to the well-nourished category (70.1%), only 27 were still at risk of malnutrition (17.5%), and 19 were still malnourished (12.3%). All the anthropometrics and functional outcomes improved from baseline to the final visit (Table 4). In particular, body weight increased by more than 3 kg and BMI increased by 1.2 units. Body composition changed with an increase in fat and body water accompanied by an increase in FFM. These changes were translated into an increase in handgrip strength by 6.9 kg on average, which means 36.5% more than at baseline. Concomitantly, the analytical parameters of nutritional status increased from baseline and inflammatory markers such as CRP as well as the ratio CRP/prealbumin were significantly reduced.

## 4. Discussion

To our knowledge, this is the first retrospective observational study showing the effect of oral nutritional supplementation along with dietary counseling and strength exercise recommendations on clinical, nutritional, and functional outcomes in outpatients attending a nutrition medical office and real-life clinical setting, including an important percentage of cancer patients under anticancer therapy who were sent to the nutrition clinical unit because of their poor nutritional status. The recommendations included both diet and exercise, which have been pointed to recently as a critical part of the clinical oncology practice by a panel of experts [24].

All patients received nutritional support according to their needs and nutritional status. The majority of patients (84.8%) were prescribed with a specialized ONS containing high protein, HMB, and vitamin D. Main outcomes included standard anthropometric, nutritional, and functional outcomes such as BIA, handgrip strength, and PG-SGA. In a subgroup of patients, it was also possible to include PhA measurement, which is considered a global prognostic factor of cellular functionality and body composition [25], and which supported some of the findings.

It is worth noting that the supplements were well tolerated in both groups and that the adherence was good or very good. Moreover, the whole program resulted in an improvement in energy and protein intake from about 1400 kcal to 2155 kcal per day, and from 0.9 to 1.5 g protein/kg BW/day on average. This is aligned with current ESPEN recommendations for older adults with or at risk of malnutrition [19].

This study also demonstrated that provision of nutritional supplementation combined with dietary counseling and strength exercise was able to provide important health benefits. Namely, it was able to improve nutritional status, shifting patients from severe malnutrition to adequate nutritional status at final visit according to PG-SGA. It was also able to increase body weight, BMI, handgrip strength, FFM, as well as nutritional biochemical markers. In fact, blood levels of albumin, prealbumin, cholesterol, and lymphocytes were brought to normal range levels, and CRP, the standard marker of inflammatory status, was reduced to normal levels. This is particularly important for improving inflammatory status in this at risk-population. The group receiving the S-ONS also had a significant reduction in CRP/prealbumin ratio, which pointed to a higher anti-inflammatory effect on this group.

Although all the patients showed improvement in main outcomes regardless of the type of ONS they received, there were some potential benefits associated to the use of the S-ONS, in particular those related to FFM and handgrip strength. This is also supported when the measurement of PhA is taken into consideration.

PhA has become an emerging parameter among bioimpedance measurements because of its value to predict impaired clinical outcomes and mortality for several diseases [22,26,27]. In addition, it is a biological marker of cellular health, as it reflects cell mass, membrane integrity, and hydration status [28]. Moreover, disease-related malnutrition and inflammation are usually associated [29]. In cancer patients and in patients with acute inflammatory processes, such as in our cohort, the progression of the disease promotes changes in inflammation and hydration that affect the accuracy of equations for predicting FFM. The data of FFM by standard BIA may be well affected by changes in hydration resulting in an overestimation of FFM or FM depending on the inflammatory process. PhA provides an integrated approach to measure body composition taking into consideration hydration and inflammatory status of patients.

Currently, there are no published randomized studies showing the impact of nutritional intervention on PhA measurements. Therefore, the present study, including observational data, provides a positive effect of nutritional intervention with ONS on PhA-related parameters.

There are some differences in PhA and associated parameters that are remarkable in the S-ONS group: significant increase in PhA, BCM, nutrition status and ICW, and concomitant decrease in the percentage of hydration and ECW compartment.

The main result of our study is an increase in PhA by 0.95 (0.13) degrees (estimated mean difference (SE)) after S-ONS treatment. This difference in PhA can be important as an increase of one degree in PhA has been associated with higher survival rates in patients in different conditions such as advanced cancer and intensive care units [27]. In addition, a recent systematic review and meta-analysis reported that, among adult patients with cancer, those with low PhA were 23% less likely to survive than those with high PhA [30]. Low PhA values were associated with higher risk of complications [31] and with higher risk of sarcopenia after adjusting for hydration abnormalities [32].

The progression of the disease induces changes in inflammation and hydration, with increase in extracellular fluids and reduction in intracellular fluids [32]. PhA is also a good marker to predict hydration status [26]. Accordingly, the reduction in ECW and increase in ICW found in the S-ONS group were positive outcomes, together with the estimated increase on average of 2.98 kg in BCM. BCM is an indicator of active cell mass (mostly muscle mass). It is not affected by hydration status and it is calculated from raw bioimpedance data and height. Therefore, it is a more reliable indicator of muscle mass than FFM.

The group of patients who received the S-ONS also improved the urinary creatinine excretion predicted from analysis of BIVA, which is a nutritional score obtained from reactance scale into creatine excretion rate.

An important contribution of our study is the high number of cancer patients that were included. There are scarce publications dealing with the use of ONS in oncology as the treatments are more focused on the disease per se than on patients’ nutritional status, primary manifested by severe muscle mass. Nutrition intervention including antioxidant supplementation, L-carnitine, or nutritional supplements with EPA combined with medication have demonstrated a significant improvement in muscle mass of patients with cancer cachexia [33,34].

Importantly, a secondary analysis of the EFFORT trial (*n* = 506) recently showed that in cancer patients with increased nutritional risk, individualized nutrition support significantly improved functional and quality-of-life outcomes and reduced mortality [35]. Cancer-related malnutrition is not only related to significant increase in mortality [36,37], but also it increases treatment interruptions [38,39], treatment toxicity [40] and decreases quality of life [41].

On the other hand, the evidence behind the use of specialized ONS containing HMB in cancer populations is very limited. Only two published studies using ONS enriched in HMB have included cancer patients. The study by De Luis et al. [42] included 43 cancer patients (29.5% of total study population), but data analysis was reported only for the total population, showing that ONS + HMB-treated patients were associated with improved body weight, nutritional status, independence levels, and quality of life. Moreover, a randomized control pilot trial conducted by Ritch et al. [43] included 61 older patients with bladder cancer undergoing radical cystectomy, and showed that ONS enriched in HMB helped preserve muscle mass in the experimental group as compared to the control group consuming a multivitamin supplement. Our study also supports the efficacy of this supplement in a larger sample population of cancer patients.

One of the limitations in the present study is that it is an observational retrospective analysis of clinical data recorded under standard of care in real-life conditions. Therefore, a cause-and-effect relationship could not be well established. Nevertheless, the use of real-world data has been considered to add additional evidence to randomized clinical trials, and to support healthcare decision makers [44]. Another limitation is that the prescription of the type of ONS was based on medical recommendations according to patient’s clinical conditions and was not based on randomization. This resulted in an unbalance of sample size and some differences on baseline characteristics of patients between both groups. These aspects were minimized by the statistical analysis which took into accounts sample weights to balance the number of patients per group using an optimal full matching procedure. In addition, including a high number of patients with cancer added heterogeneity to the study sample with regards to muscle related outcomes, which may have been influenced by the anti-cancer treatment. The irregularity on follow-up visit schedule may influence the results.

Nevertheless, this study shows that a program including dietary and strength exercise counseling together with the use ONS is beneficial to support nutritional, clinical, and functional outcomes, especially those related to improved muscle health in adult outpatients with or at risk of malnutrition. In addition to this, the increase of 0.95 grades in PhA and 2.98 kg in BCM, that was found in the S-ONS group, were very important findings and support the effectiveness of the use of this supplement in the recovery of muscle mass, which serves as a global disease prognosis in malnourished patients. This study also adds more evidence to the use of an ONS, high in protein and with HMB, to improve nutritional status and body composition in malnourished patients, especially in cancer patients. However, more studies are needed to confirm these results.

## Figures and Tables

**Figure 1 nutrients-13-04355-f001:**
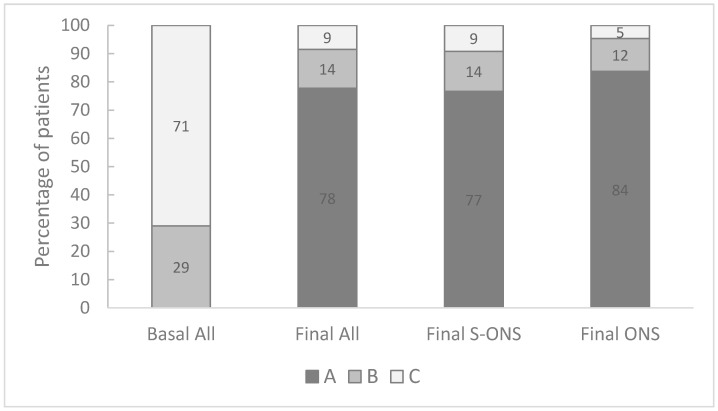
Nutritional status according to patient-generated subjected global assessment tool (PG-SGA): A, well-nourished; B, moderately or suspected of being malnourished; C, severely mal-nourished.

**Figure 2 nutrients-13-04355-f002:**
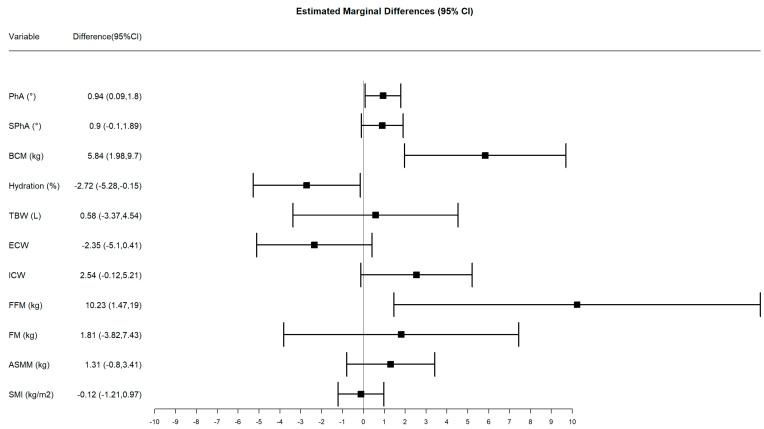
Estimated marginal differences between groups at final visit and 95% confidence intervals obtained by the adjusted linear regression model. Confidence intervals that do not overlap with zero denote significant differences. TBW: total body water; ECW: extracellular water; ICW: intracellular water; FFM: fat-free mass; FM: fat mass; BCM: body cell mass; ASMM: appendicular skeletal muscle mass; SMI: skeletal muscle index; PhA: phase angle; SPhA: standardized phase angle.

**Table 1 nutrients-13-04355-t001:** Baseline characteristics of study participants.

	Overall (N = 283)	S-ONS N = 240	ONS N = 43	*p*-Value (between Groups)
Age (years)	60.9 (14.2)	59.0 (14.6)	62.7 (13.8)	0.131
Gender N (%)				0.107
Male	149 (53%)	121 (50.4)	28 (65.1)	
Female	134 (47%)	119 (49.6)	15 (34.9)	
Admission				0.304
Oncology surgery	179 (63%)	155 (64.6)	24 (55.8)	
General surgery	44 (16%)	34 (14.2)	10 (23.3)	
Others	60 (21%)	51 (21.2)	9 (20.9)	
Anthropometrics				
Basal weight (kg)	63.1 (12.9)	63.2 (12.9)	62.6 (13.3)	0.770
Basal BMI (kg/m^2^)	23.2 (4.4)	23.2 (4.5)	23.1 (4.3)	0.827
Biceps fold (mm)	10.4 (4.8)	10.7 (5.0)	9.0 (2.6)	0.026
Arm circumference (cm)	25.5 (3.9)	25.7 (3.9)	24.1 (3.5)	0.011
Handgrip strength				
Dynamometry (kg)	18.7 (13.0)	18.3 (13.0)	20.4 (12.6)	0.333
Body composition				
Fat mass (kg)	15.3 (7.7)	15.5 (7.9)	14.5 (6.5)	0.438
Fat-free mass (kg)	47.9 (9.1)	47.9 (9.1)	48.1 (9.3)	0.859
Total body water (kg)	35.1 (6.6)	35.1 (6.6)	35.1 (6.8)	0.950
Nutritional status				
PG-SGA				0.704
At risk of malnutrition	82 (29.0)	68 (28.3)	14 (32.6)	
Malnutrition	201 (71.0)	172 (71.7)	29 (67.4)	
SEDOM-SENPE Calorie malnutrition Protein malnutrition Protein calorie malnutrition				0.074
	20 (7.1)	20 (8.3)		
	16 (5.7)	15 (6.2)	1 (2.3)	
	247 (87.3)	205 (85.4)	42 (97.7)	
Biochemistry				
Albumin (g/dL)	2.9 (0.7)	2.9 (0.6)	2.9 (0.6)	0.895
Prealbumin (mg/dL)	19.2 (6.5)	18.9 (6.4)	20.7 (6.8)	0.088
CRP (mg/L)	32.2 (51.0)	32.6 (50.9	30.2 (51.7)	0.772
CRP/Prealbumin	0.24 (0.50)	0.25 (0.52)	0.18 (0.33)	0.395
Cholesterol (mg/dL)	150.1 (41.3)	151.1 (40.7)	144.5 (44.5)	0.335
Lymphocytes (mm^3^ × 10^−3^)	1.5 (1.0)	1.5 (1.1)	1.4 (0.5)	0.346

Data are mean (SD). ONS: standard oral nutritional supplement; S-ONS: specialized oral nutritional supplement; BMI: body mass index; PG-SGA: patient-generated subjective global assessment of nutritional status. SENPE–SEDOM: malnutrition categories according to the Spanish Society of Parenteral and Enteral Nutrition (SENPE) and the Spanish Society of Medical Documents (SEDOM). CRP: C reactive protein.

**Table 2 nutrients-13-04355-t002:** Body composition, dynamometry, and laboratory outcomes at baseline and after receiving oral nutritional supplements.

	S-ONS N = 240		ONS N = 43	
	Basal	Final	Basal	Final
Anthropometrics				
Body weight (kg)	63.2 (12.9)	67.2 (13.5) *	62.6 (13.3)	66.2 (14.6) *
BMI (kg/m^2^)	23.2 (4.5)	24.7 (4.8) *	23.1 (4.3)	24.4 (4.8) *
Biceps fold (mm)	10.7 (5.0) #	12.2 (5.3) *	9.0 (2.6)	10.4 (3.6) *
Arm circumference (cm)	25.7 (3.9) #	27.5 (4.1) *#	24.1 (3.5)	25.7 (4.3) *
Handgrip strength				
Dynamometry (kg)	18.3 (13.0)	25.2 (14.4) *	20.4 (12.6)	24.9 (14.7) *
Body composition				
Fat mass (kg)	15.5 (7.9)	18.1 (9.0) *	14.5 (6.5)	17.0 (8.4) *
Fat-free mass (kg)	47.9 (9.1)	49.1 (9.0) *	48.1 (9.3)	49.0 (8.8)
Total body water (kg)	35.1 (6.6)	35.9 (6.5) *	35.1 (6.8)	36.0 (6.6) *
Biochemistry				
Albumin (g/dL)	2.9 (0.7)	3.8 (0.5) *	2.9 (0.6)	3.6 (0.4) *
Prealbumin (mg/dL)	18.9 (6.4)	24.7 (6.7) *	20.7 (6.8)	24.4 (7.1) *
CRP (mg/L)	32.6 (50.9)	10.8 (34.0) *	30.2 (51.7)	9.3 (29.9) *
CRP/Prealbumin	0.25 (0.52)	0.08 (0.40) *	0.18 (0.33)	0.15 (0.82)
Cholesterol (mg/dL)	151.1 (40.7)	178.7 (48.4) *	144.5 (44.5)	165.1 (36.9) *
Lymphocytes (mm^3^ × 10^−3^)	1.5 (1.1)	1.9 (0.9) *	1.4 (0.5)	1.8 (0.8) *

Data are mean (SD). ONS: standard oral nutritional supplement; S-ONS: specialized oral nutritional supplement; BMI: body mass index; CRP: C reactive protein. *: *p* < 0.05 vs. Basal. #: *p* < 0.05 vs. ONS obtained from the adjusted model.

**Table 3 nutrients-13-04355-t003:** Anthropometric, body composition, dynamometry, and laboratory outcomes at baseline and after receiving oral nutritional supplements for the subgroup analysis by phase angle.

	S-ONS N = 31		ONS N = 12	
	Basal	Final	Basal	Final
Anthropometrics				
Body weight (kg)	60.5 (10.8)	63.9 (11.5) *	58.9 (11.7)	60.5 (12.5)
BMI (kg/m^2^)	22.0 (3.2)	23.2 (3.5) *	22.2 (2.9)	22.8 (3.3)
Biceps fold (mm)	9.0 (3.2)	10.3 (3.6) *	8.1 (1.5)	9.1 (2.0) *
Arm circumference (cm)	23.7 (2.6)	25.2 (2.8) *	22.9 (2.6)	23.9 (3.2) *
Handgrip strength				
Dynamometry (kg)	24.7 (11.3)	30.0 (11.0) *	18.6 (12.7)	23.2 (14.7) *
Body composition by phase angle				
PhA (°)	5.2 (1.1)	6.0 (1.1) *	4.9 (1.0)	5.2 (1.0)
SPhA (°)	−0.9 (1.1)	−0.1 (0.9) *	0.2 (1.7)	0.6 (1.7)
BCM (kg)	24.2 (6.7)	26.8 (5.9) *#	22.1 (5.2)	22.5 (3.7)
Nutritional status (mg/m/24 h)	728.5 (200.8)	807.3 (180.7) *	687.5 (145.4)	717.5 (126.9)
Hydration (%)	74.8 (3.3)	72.9 (1.3) *	75.5 (4.4)	74.8 (4.3)
TBW (L)	36.6 (6.6)	36.5 (6.6)	35.4 (8.6)	36.2 (7.8)
ECW (L)	18.0 (3.2)	16.7 (3.1) *	19.0 (5.4)	18.4 (5.6)
ICW (L)	18.3 (4.7)	19.8 (4.4) *	17.3 (3.4)	17.8 (3.1)
FFM (kg)	49.0 (8.8) #	49.5 (9.5) #	36.1 (17.7)	41.0 (15.7) *
FM (kg)	11.4 (5.5) #	13.1 (6.5)	9.0 (6.9)	12.1 (7.0)
ASMM (kg)	18.3 (4.2)	18.7 (4.0) *	17.3 (3.7)	17.5 (3.9)
SMI (kg/m^2^)	8.7 (1.7)	8.5 (1.7)	8.8 (1.6)	8.6 (1.8)
Biochemistry				
Albumin (g/dL)	3.2 (0.6)	3.9 (0.6) *	3.1 (0.4)	3.6 (0.4)
Prealbumin (mg/dL)	19.4 (5.5)	24.9 (7.0) *	18.8 (4.7)	21.8 (5.7)
CRP (mg/L)	11.2 (17.2)	6.6 (19.2)	20.8 (41.0)	6.2 (5.6)
CRP/Prealbumin	0.07 (0.14)	0.06 (0.24)	0.13 (0.27)	0.03 (0.04)
Cholesterol (mg/dL)	155.5 (39.4)	178.9 (42.0) *	163.6 (46.0)	178.4 (41.4)
Lymphocytes (mm^3^ × 10^−3^)	1.6 (0.5)	1.5 (0.4) *	2.0 (0.6)	1.9 (1.0) *

Data are mean (SD). ONS: standard oral nutritional supplement; S-ONS: specialized oral nutritional supplement; BMI: body mass index; CRP: C reactive protein; TBW: total body water; ECW: extracellular water; ICW: intracellular water; FFM: fat-free mass; FM: fat mass; BCM: body cell mass; ASMM: appendicular skeletal muscle mass; SMI: skeletal muscle index; PhA: phase angle; SPhA: standardized phase angle. *: *p* < 0.05 vs. Basal. #: *p* < 0.05 vs. ONS obtained from the adjusted model.

**Table 4 nutrients-13-04355-t004:** Body composition, dynamometry, and laboratory outcomes in oncological patients (N = 155) at baseline and after receiving a high-protein oral nutritional supplement with HMB.

	Basal	Final
	Anthropometrics	
Body weight (kg)	63.1 (12.4)	66.2 (12.4) *
BMI (kg/m^2^)	23.3 (4.3)	24.5 (4.5) *
Biceps fold (mm)	10.9 (5.2)	12.2 (5.4) *
Arm circumference (cm)	25.5 (3.8)	27.1 (3.9) *
	Handgrip strength	
Dynamometry (kg)	18.9 (12.9)	25.8 (14.4) *
	Body composition	
Fat mass (kg)	15.6 (7.8)	17.6 (8.3) *
Fat-Free mass (kg)	47.7 (8.9)	48.7 (8.6) *
Corporal water (kg)	34.9 (6.5)	35.6 (6.3) *
	Biochemistry	
Albumin (g/dL)	3.0 (0.7)	3.8 (0.5) *
Prealbumin (mg/dL)	18.3 (5.9)	24.0 (7.1) *
CRP (mg/L)	31.7 (51.9)	13.9 (41.9) *
CRP/Prealbumin	0.27 (0.59)	0.12 (0.49) *
Cholesterol (mg/dL)	157.0 (39.2)	181.6 (43.7) *
Lymphocytes (mm^3^ × 10^−3^)	1.5 (1.2)	1.9 (1.0) *

Data are mean (SD). BMI: body mass index; CRP: C reactive protein. *: *p* < 0.05 vs. Basal.

## Data Availability

The data that support the findings of this study are available from the corresponding author, upon reasonable request.

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
