# Peer review of "Effect on an Oral Nutritional Supplement with β-Hydroxy-β-methylbutyrate and Vitamin D on Morphofunctional Aspects, Body Composition, and Phase Angle in Malnourished Patients"

_nutrients, 2021, doi:10.3390/nu13124355_

Round 1

Reviewer 1 Report

This work investigated the effect of a high calorie high protein oral nutritional supplement (ONS) with HMB and vitamin D on nutritional status, body weight, and muscle- related outcomes in adult patients with or at risk of malnutrition under the standard care. This study is valuable because it shows that disease-related malnutrition and muscle loss associated with it are not inevitable when a nutritional intervention is administered. In addition, it’s visible that regardless of the type of ONS, their use by patients helps to improve nutritional, clinical and functional outcomes.

To improve the manuscript, please:

1) add to section Materials and Methods (2.1. Study design) additional information regarding recommendations for muscle strengthening. What exactly did these recommendations contain - did the dietitian recommend any specific strength exercises?  If so, at what frequency? Was ONS recommended to be drunk after exercise?  It is well known that the combination of ONS supplementation with exercise gives the best effect, but the description of the study does not indicate the importance of exercises. Perhaps it would be worth addressing this point in study limitation section.

2) add a detailed description of the muscle strength measurement in the Materials and Methods section (2.5. Evaluation of Nutritional status). Please add information whether both hands have been tested; how many times the measurement was carried out and what cut-off points were used.

3) I would like to point out that in several places in the manuscript there is a comma instead of a dot:

- line 249 (1,4 kg/m2)

- line 223 (0,9 g of protein)

- line 227 (1,5 g/kg body weight)

- line 306 (12,3%)

- line 310 (6,9 kg on average)

Author Response

Dear Editor and Reviewers,

We would like to thank you very much for your constructive comments and suggestions which have undoubtedly helped us to improve our manuscript.

We have taken these comments and suggestions into consideration and have revised the paper accordingly. We have made all possible efforts to respond to each of the reviewers’ comments and have edited the manuscript where we were able to address the reviewers’ suggestions fully.

We have provided the replies to the comments in the following section and have made the changes with the Track Changes option. In the next section, you will find detailed answer to each reviewer’s comment and the lines in the manuscript where the changes were done.

We hope that our revised manuscript may now be found acceptable for publication in the journal. Nevertheless, we are of course willing to revise it further according to any other suggestions or concerns raised by the Editor or the Reviewers.

Yours faithfully,

Isabel Cornejo-Pareja

Reviewer 2 Report

This is a retrospective study of data from clinical practice to observe the effect of a high calorie high protein oral nutritional supplement  with HMB on nutritional status in adults with or at risk of malnutrition.

  1. The first thing that stands out is the huge disproportion between the two study groups. The control group is about 6 times smaller than the study group. It could be due to BIAS. How do the authors explain these differences?
  2. The data on the prescribed diet are very general. It seems that the dietician was little involved in the nutritional management of the patient.
  3. It would be essential to add a control group following a diet without supplements. This would allow a proper evaluation of the real effectiveness of the two different types of supplements. 
  4. what are the nutritional characteristics of the supplement administered in the control group? it would be useful to add a table in which it would be clear what the nutritional differences are between the two supplements apart from the presence of HMB.
  5. Since the study was sponsored by Abbott the authors should specify the name of the product used. "Ensure"?
  6. Beware of some plagiarised paragraphs between methods

Author Response

(The authors gave the same response as above.)

Round 2

Reviewer 2 Report

The authors have made some improvements to the paper. Structural shortcomings remain.The numbers for a retrospective study are too small especially in the group where subgroup analysis by Phase Angle was carried out. I think that no realistic conclusions can be drawn about the usefulness of S-ONS in this type of patients. It appears that dietary intervention had a big impact while supplementation had much less.
I suggest that the authors prepare a new study with larger numbers with two groups: one without supplementation and one with supplementation. S-ONS with these numbers cannot be evaluated.